# Membrane Interactivity of Capsaicin Antagonized by Capsazepine

**DOI:** 10.3390/ijms23073971

**Published:** 2022-04-02

**Authors:** Maki Mizogami, Hironori Tsuchiya

**Affiliations:** 1Department of Anesthesiology, Central Japan International Medical Center, Minokamo 505-8510, Gifu, Japan; m-mizogami@cjimc-hp.jp; 2Department of Dental Basic Education, School of Dentistry, Asahi University, Mizuho 501-0296, Gifu, Japan

**Keywords:** capsaicin, membrane interactivity, biomimetic membrane, capsazepine, competitive antagonism

## Abstract

Although the pharmacological activity of capsaicin has been explained by its specific binding to transient receptor potential vanilloid type 1, the amphiphilic structure of capsaicin may enable it to act on lipid bilayers. From a mechanistic point of view, we investigated whether capsaicin and its antagonist capsazepine interact with biomimetic membranes, and how capsazepine influences the membrane effect of capsaicin. Liposomal phospholipid membranes and neuro-mimetic membranes were prepared with 1,2-dipalmitoylphosphatidylcholine and with 1-palmitoyl-2-oleoylphosphatidylcholine and sphingomyelin plus cholesterol, respectively. These membrane preparations were subjected to reactions with capsaicin and capsazepine at 0.5–250 μM, followed by measuring fluorescence polarization to determine the membrane interactivity to modify the fluidity of membranes. Both compounds acted on 1,2-dipalmitoylphosphatidylcholine bilayers and changed membrane fluidity. Capsaicin concentration-dependently interacted with neuro-mimetic membranes to increase their fluidity at low micromolar concentrations, whereas capsazepine inversely decreased the membrane fluidity. When used in combination, capsazepine inhibited the effect of capsaicin on neuro-mimetic membranes. In addition to the direct action on transmembrane ion channels, capsaicin and capsazepine share membrane interactivity, but capsazepine is likely to competitively antagonize capsaicin’s interaction with neuro-mimetic membranes at pharmacokinetically-relevant concentrations. The structure-specific membrane interactivity may be partly responsible for the analgesic effect of capsaicin.

## 1. Introduction

Capsaicin, a pungent component of chili peppers belonging to the genus *Capsicum*, binds to transient receptor potential vanilloid type 1 (TRPV1), which is a ligand-gated nonselective cation channel expressed in peripheral nociceptive neurons, dorsal root ganglia and trigeminal ganglia, and also in neuronal cells of the central nervous system. Since TRPV1 is involved in the transmission and modulation of pain, TRPV1 agonist capsaicin produces a burning sensation immediately after application and pain by stimulating nociceptive neurons, whereas paradoxically it can relieve pain. Capsaicin is not only used for the perioperative pain management, but also clinically expected as an agent for multimodal analgesia in addition to application to various neuropathic chronic pain conditions [1,2]. Capsaicin has been also suggested to improve local anesthesia, especially in inflamed tissues, by combining with local anesthetics [3].

The pharmacological activity of capsaicin is explained by its specific action on TRPV1 ion channels [4,5]. After exposure to a high or repeated dose of capsaicin, TRPV1 is desensitized with the subsequent inhibition resulting in analgesia. The direct interaction with TRPV1 has been referred to as one of mechanisms underlying the analgesic effect of capsaicin, while not all of the pharmacological actions of capsaicin and its antagonist are dependent on TRPV-1 [6]. Capsaicin has an amphiphilic structure (Figure 1), which may enable capsaicin to act on lipid bilayers. In addition to specific binding to target proteins, amphiphiles, including drugs and phytochemicals, can interact with lipid bilayers to affect membrane fluidity, permeability, elasticity and curvature [7,8], and membrane lipid phase transition and lipid bilayer thickness [8,9]. Such membrane interactions modulate the functions of membrane-embedded channels through protein conformational changes [10], mechanistically contributing to the pharmacological effects of amphiphilic anesthetics, analgesics, anti-inflammatory drugs and others [11,12,13].

Startek et al. [14] hypothesized that transient receptor potential (TRP) ion channels could be functionally regulated by their surrounding membrane lipid environment. Subsequently, they reported that mouse TRP function and membrane localization are modulated by interactions with a specific membrane lipid [15]. As suggested by Hanson et al. [16], the membrane environment is likely to play a crucial role in mediating the effect of capsaicin because capsaicin has the structure to interact with lipid membranes, and TRPV1 is an integral transmembrane protein. Membrane fluidity or thickness changes influence sensory TRP channel structure and gating [17]. Different studies reported the membrane interaction of capsaicin to modify membrane fluidity, permeability and phase transition [18,19,20,21], in which capsaicin was subjected to the reactions with model membranes consisting of a single species of phospholipid or a mixture of phosphatidylcholine and cholesterol, followed by fluorescence polarization analysis, differential scanning calorimetry and nuclear magnetic resonance spectroscopy. Although the composition of membrane-constituting lipids varies by the type of cells, the membrane interactivity of capsaicin has not been characterized on neuronal membranes.

A synthetic analogue of capsaicin, capsazepine (Figure 1), inhibits the effect of capsaicin on nociceptive neurons of rats [22], reduces the current response to capsaicin in voltage-clamped dorsal root ganglion neurons of rats [23], and prevents the antinociceptive effect of capsaicin systematically administered to rats [24]. The pharmacological mechanisms for these inhibitory effects have been interpreted by capsazepine’s antagonistic action on TRPV1 [25,26]. Despite having the amphiphilic structure, however, the property of capsazepine to interact with lipid bilayers and the antagonistic relationship in membrane interaction between capsaicin and capsazepine remain to be elucidated.

We investigated whether capsaicin and capsazepine interact with biomimetic membranes and how capsazepine influences the membrane effect of capsaicin when used in combination. In the present study, protein-free lipid bilayer membranes were used to focus on the interactions of these compounds with membrane lipid components. Here, we report that capsaicin and capsazepine share the membrane interactivity, but capsazepine competitively inhibits the effect of capsaicin on neuro-mimetic membranes.

## 2. Results

### 2.1. Interaction with Phosphatidylcholine Liposomal Membranes

Both capsaicin and capsazepine interacted with liposomal membranes consisting of 1,2-dipalmitoylphosphatidylcholine (DPPC) to induce significant changes in membrane fluidity. Capsaicin increased the fluidity of DPPC membranes at 50–250 μM, as shown by decreases of fluorescence polarization measured with fluorescent probe diphenyl-1,3,5-hexatriene (DPH) (Figure 2a), whereas capsazepine decreased the fluidity of DPPC membranes, as shown by increases of DPH fluorescence polarization (Figure 2b).

### 2.2. Interaction with Neuro-Mimetic Membranes

Capsaicin interacted with neuro-mimetic membranes, which were prepared with 1-palmitoyl-2-oleoylphosphatidylcholine (POPC) and sphingomyelin plus cholesterol, resulting in concentration-dependent changes of membrane fluidity (Figure 3a). Capsaicin showed biphasic effects on neuro-mimetic membranes to increase their fluidity at 0.5–200 μM, but decrease fluidity at concentrations higher than 250 μM. In contrast to capsaicin, capsazepine decreased the fluidity of neuro-mimetic membranes at 10–250 μM (Figure 3b).

The membrane interactivity of capsaicin and capsazepine varied by a difference in membrane lipid composition, as shown by relative changes of DPH polarization (Table 1). Capsaicin interacted with neuro-mimetic membranes at 50 μM more potently than DPPC membranes, but less potently at higher concentrations. Capsazepine induced larger fluidity decreases in DPPC membranes than in neuro-mimetic membranes.

### 2.3. Effect of Capsaicin on Neuro-Mimetic Membranes and Inhibition by Capsazepine

The membrane effects of capsaicin were compared between neuro-mimetic membranes pretreated with and without capsazepine (Figure 4). Capsaicin interacted with neuro-mimetic membranes to increase their fluidity at 10 and 50 μM in the absence of capsazepine. However, its membrane effects were significantly inhibited by pretreating the membranes with 10, 50 and 100 μM capsazepine.

### 2.4. Membrane Interaction Antagonism between Capsaicin and Capsazepine

Concentration-response curves in neuro-mimetic membrane interaction were prepared for capsaicin of 0.5–200 μM in the presence or absence of capsazepine of 50 μM (Figure 5). Capsazepine antagonized the membrane-fluidizing effect of capsaicin. The effect of a competitive antagonist can be overcome by increasing the dose of an agonist; the dose-response curve of an agonist is shifted to the right with the degree of rightward shift relating to the antagonist’s receptor affinity [27]. In the interaction with neuro-mimetic membranes, a rightward shift of the concentration-polarization increase curve was obtained almost without affecting the maximum response to capsaicin. Assuming capsaicin and capsazepine act on the same site in membrane lipid bilayers, on the analogy of drug–receptor interaction, capsazepine is likely to exhibit the membrane interactivity like a competitive antagonist. For capsazepine, calculated values of IC_50_ (a concentration to inhibit the membrane effect by 50%) and pA2 (an estimate of the antagonist’s potency) were 38.8 μM and 4.44, respectively.

## 3. Discussion

We have comparatively studied the biomimetic membrane interactivity of TRPV1 agonist capsaicin and antagonist capsazepine. Our main findings are as follows: (1) both compounds interact with phospholipid bilayers and modify membrane properties depending on membrane lipid composition; (2) capsaicin concentration-dependently increases the fluidity of neuro-mimetic membranes at low micromolar concentrations; (3) capsazepine decreases the fluidity of neuro-mimetic membranes to inhibit the membrane effect of capsaicin; and (4) capsazepine competitively antagonizes the neuro-mimetic membrane interactivity of capsaicin. Taken together, it is considered that capsaicin and capsazepine interact with lipid bilayers consisting of phospholipids and cholesterol to change their membrane fluidity differently, and antagonism between capsaicin and capsazepine occurs on interaction not only with TRPV1 channels, but also with neuro-mimetic membranes.

Capsaicin is rapidly absorbed from the gastrointestinal tracts after oral administration and robustly absorbed from the skin after topical administration. In pharmacokinetic studies, human subjects and rats were orally administered with 5 g of capsicum (corresponding to 87 μmol capsaicin) [28] and capsaicin of 30 mg/kg [29], respectively, followed by determination of capsaicin in blood. The peak plasma concentration of capsaicin was found to be sub-micromolar to low micromolar levels. At pharmacokinetically-relevant concentrations and the concentrations commonly used in clinical studies [7], capsaicin has been confirmed to interact with neuro-mimetic membranes in the present experiment.

Capsaicin has the structure composed of 4-hydroxy-3-methoxyphenyl or vanillyl group, amide bond and 8-methyl-7-nonenal moiety (Figure 1). With respect to its presence in membrane lipid bilayers, Aranda et al. [18] and Tsuchiya [19] suggested that a hydrophobic tail is aligned along the acyl chains of phospholipids, whereas hydroxyl and amide groups are located near the lipid/water interface to form hydrogen bonding with the polar head groups of phospholipids. Swain and Kumar Mishra [30] conducted a fluorescence quenching study with cetylpyridinium chloride to determine the location of capsaicin in phospholipid bilayer membranes. Their results indicated that a hydrophobic tail and a phenolic group are present inside the hydrophobic core region and near the head group region of lipid bilayers, respectively. In molecular dynamics simulations, Hanson et al. [16] demonstrated that the distribution of capsaicin in lipid bilayers aligns the substituted aromatic ring with the carbonyl moiety of POPC. In such membrane lipid bilayers, capsaicin molecules could flip from the extracellular to the intracellular leaflet and subsequently access the intracellular TRPV1 binding site.

For the membrane effects of capsaicin at relatively low concentrations, we speculate that capsaicin is intercalated between phospholipid molecules to perturb the packing of membrane lipids, resulting in an increase of membrane fluidity. Aranda et al. [18] reported that capsaicin shifts the phase transition temperature of membrane DPPC to lower values, but to higher values at relatively high concentrations. In unilamellar vesicles composed of phospholipids or phospholipids plus cholesterol, the membrane fluidity is liquid-ordered (*L_o_*) ≈ gel (*L**_β_*) < liquid-disordered (*L_d_*) < liquid-crystalline (*L**_α_*) phase [31]. Ingólfsson et al. [32] investigated partition of five amphiphilic phytochemicals (including capsaicin, curcumin, (−)-epigallocatechin-3-gallate (EGCG), genistein and resveratrol) into POPC bilayer membranes and their effects in altering the membrane properties. They concluded that capsaicin reaches more deeply into the hydrophobic region than other phytochemicals, and many effects of all the tested phytochemicals are due to their membrane perturbation. However, it was recently indicated that capsaicin and curcumin increase membrane fluidity, whereas EGCG, genistein and resveratrol inversely decrease [8]. The present study has confirmed that capsaicin increases the fluidity of neuro-mimetic membranes at low micromolar concentrations but decreases at higher concentrations. The membrane-rigidifying effect may result from the interdigitation of phospholipid acyl chains that capsaicin induces with increasing concentrations [19].

Membrane-active drugs exhibit the membrane interactivity that varies by a difference in their structures and can be discriminated even between stereoisomers [33,34]. The comparative membrane interaction study of capsaicinoids showed that capsaicin’s structural analog *N*-vanillylnonanamide increases the fluidity of POPC membranes at low micromolar concentrations, but less potently than capsaicin [19]. In the present study, capsaicin (8-methyl-*N*-vanillyl-*trans*-6-nonenamide) increased the fluidity of DPPC membranes at ~250 μM and neuro-mimetic membranes at as low concentration as 2 μM, whereas capsazepine (*N*-[2-(4-chlorophenyl)ethyl]-1,3,4,5-tetrahydro-7,8-dihydroxy-2*H*-2-benzazepine-2-carbothioamide) decreased the fluidity of DPPC membranes and neuro-mimetic membranes, and inhibited the effect of capsaicin on neuro-mimetic membranes with an IC_50_ of 38.8 μM. Since capsaicinoids (capsaicin and *N*-vanillylnonanamide) with a vanillyl group commonly fluidized the membranes but not capsazepine, the 1,3,4,5-tetrahydro-7,8-dihydroxybenzazepine structure specific to capsazepine may be responsible for its membrane-rigidifying effect.

While fluorescent probe DPH and 1-(4-trimethylammoniumphenyl)-6-phenyl-1,3,5-hexatriene (TMA-DPH) have been used to determine physicochemical properties of phospholipid and phospholipid/cholesterol bilayers, DPH penetrates into lipid bilayers deeply, but TMA-DPH has a more superficial location than DPH because of its additional charged group [35]. When interacting with POPC liposomal membranes, capsaicin-induced decreases of DPH polarization were greater than those of TMA-DPH polarization [19], suggesting that capsaicin preferentially acts on the membrane regions consisting of phospholipid acyl chains to increase their fluidity. Both capsaicin and capsazepine significantly changed (decreased or increased) DPH polarization of DPPC and neuro-mimetic membranes at low micromolar concentrations in the present study. It is speculated that capsaicin and capsazepine act on the same site in membrane lipid bilayers (hydrophobic region) by inserting an 8-methyl-6-nonenamide moiety and a 4-chlorophenylethyl-2-carbothioamide moiety into the phospholipid acyl chains, respectively. The membrane-fluidizing effects of capsaicin are understandable given the fact that alkyl compounds increase the fluidity of phospholipid membranes with the potency depending on an increase of the alkyl chain length [36]. For capsazepine, a chlorophenyl group at the chain terminal may be associated with its membrane-rigidifying effects. A 4-hydroxy-3-methoxyphenyl moiety (vanillyl group) of capsaicin and a 7,8-dihydroxybenzazepine moiety of capsazepine can be anchored in the phospholipid head group of the membrane surface. Such a bulky structural moiety of capsazepine may reach up to the upper acyl chain regions, possibly affecting their fluidity.

When administered to a rat tail and spinal cord, capsazepine antagonized the nociceptive responses to capsaicin with IC_50_ of 230–254 nM, and when subcutaneously injected to rats, capsazepine (100 μmol/kg) antagonized the C-fiber responses evoked by the systemic administration of capsaicin (20 μmol/kg) [22]. Capsazepine, at 10 μM, reduced the current responses to 500 nM capsaicin of voltage-clamped dorsal root ganglion neurons of rats [23]. Capsazepine subcutaneously injected to mice and rats at 10–100 μmol/kg reversed the antinociceptive action of capsaicin systemically administered at 20 μmol/kg [24].

Besides antagonizing the effects of capsaicin on TRPV1, capsazepine has been confirmed to antagonize the interaction of capsaicin with neuro-mimetic membranes. Amphiphilic drugs including capsaicin can modulate the functions of membrane-embedded channels by altering the intrinsic properties of lipid bilayers. Lundbæk et al. [7] investigated whether amphiphiles could regulate the functions of membrane channels by acting on membrane lipid bilayers. Structurally different amphiphilic compounds commonly decreased lipid bilayer stiffness and promoted inactivation of transmembrane voltage-dependent sodium channels. Among them, capsaicin and capsazepine produced similar changes in elasticity of the lipid bilayers that consisted of either 1,2-dioleoylphosphatidylcholine (DOPC) or diphytanoylphosphatidylcholine and *n*-decane. In contrast, capsaicin and capsazepine produced opposite changes in fluidity of the neuro-mimetic membranes that were prepared with POPC and specific neuronal lipid sphingomyelin plus cholesterol. Sharma et al. [21] revealed that capsaicin increases the fluidity of DOPC membranes, but decreases the fluidity of membranes prepared with DOPC plus cholesterol. Since the membrane effect of capsaicin varies depending on membrane lipid composition [19,21], membrane cholesterol may participate in characterizing the membrane interactivity of capsaicin. Prakash and Srinivasan [37] assessed the influences of dietary spices and bioactive spice constituents including capsaicin on the fluidity and lipid composition of intestinal brush border membranes of rats. Capsaicin increased membrane fluidity in the ileum and jejunum, but decreased fluidity in the duodenum, which was concordant with changes in membrane lipid composition (cholesterol/phospholipid ratio). Liu et al. [38] examined the importance of membrane cholesterol in the function and expression of TRPV1 in rat dorsal ganglion neurons by experimentally depleting cholesterol with methyl-β-cyclodextrin. Their results showed that the cholesterol level determines the activity and amount of membrane TRPV1, suggesting the localization of TRPV1 in cholesterol-rich microdomains or lipid rafts [15]. Cholesterol would influence the activity of transmembrane channels by directly interacting with ion channels, modifying the physicochemical properties of membranes, and/or facilitating the formation of membrane microdomains [14]. TRP channels present in cholesterol-concentrated lipid rafts are modulated by changes in lipid bilayer fluidity, thickness and phase [13,14].

As far as we know, this is the first report on the antagonistic membrane interaction between capsaicin and capsazepine. In addition to the direct action on transmembrane TRPV1 channels, the structure-specific membrane interactivity may be partly responsible for the analgesic effect of capsaicin. Our findings would give a novel insight into the pharmacological mechanism of amphiphilic TRPV1 agonist and competitive antagonist.

## 4. Materials and Methods

### 4.1. Chemicals

Capsaicin and capsazepine were obtained from Funakoshi (Osaka, Japan). DPPC, POPC and porcine brain sphingomyelin were purchased from Avanti Polar Lipids (Alabaster, AL, USA), cholesterol from Wako Pure Chemicals (Osaka, Japan), and DPH from Molecular Probes (Eugene, OR, USA). Dimethyl sulfoxide (DMSO) and ethanol of spectroscopic grade (Kishida; Osaka, Japan) and water of liquid chromatographic grade (Kishida) were used for preparing reagent solutions. All other chemicals were of the highest analytical grade available commercially.

### 4.2. Membrane Preparation

Liposomal membranes were prepared by the ethanol injection method [39], in which a lipid solution of ethanol is rapidly injected to an excess of buffer to form unilamellar vesicles, as reported by Okimoto et al. [40]. In brief, the dry film of phospholipids and cholesterol was dissolved with an ethanolic solution of DPH. An aliquot (250 µL) of the resulting solution (total lipids of 10 mM and DPH of 50 µM) was injected four times into 199 mL of 10 mM 4-(2-hydroxyethyl)-1-piperazineethanesulfonic acid buffer (pH 7.4, containing 125 mM NaCl and 25 mM KCl) under stirring above the phase transition temperatures of phospholipids. The membrane lipid compositions were 100 mol% DPPC for DPPC membranes and 45 mol% POPC, 10 mol% sphingomyelin and 45 mol% cholesterol for neuro-mimetic membranes [19,34]. The liposomal particle size was presumed to be approximately 100–400 nm, with reference to experimental data on liposome preparation by means of the ethanol injection method [41].

### 4.3. Determination of Membrane Interactivity

Capsaicin and capsazepine were dissolved in DMSO. An aliquot of the resulting solution was applied to the DPH-labelled DPPC membranes and DPH-labelled neuro-mimetic membranes so that final concentrations ranged from 0.5–250 μM for capsaicin and capsazepine. The concentration of DMSO was adjusted to be 0.5% (*v*/*v*) of the total volume so as not to affect the fluidity of intact membranes. Control experiments were conducted with the addition of an equivalent volume of vehicle DMSO. When investigating capsaicin’s membrane effects inhibited by capsazepine, neuro-mimetic membranes were pretreated with 0–100 μM capsazepine at 37 °C for 15 min, and then treated with 10 or 50 μM capsaicin. After all membrane samples were subjected to the reactions at 37 °C for 45 min, DPH fluorescence polarization was measured at 360 nm for the excitation wavelength and 430 nm for the emission wavelength by an FP-777 spectrofluorometer (Japan Spectroscopic Cooperation; Tokyo, Japan), equipped with a polarizer and a cuvette thermo-controlled at 37 °C. Polarization values were calculated by the formula (*I*_VV_ − *GI*_VH_)/(*I*_VV_ + *GI*_VH_) according to the method of Ushijima et al. [42], in which *I* is the fluorescence intensity and the subscripts V and H refer to the vertical and horizontal orientation of excitation and emission polarizer, respectively. The grating correction factor (*G* = *I*_HV_/*I*_HH_) is the ratio of the detection system sensitivity for vertically and horizontally polarized light, which was used to correct the polarizing effects of a monochromator. Compared with controls, a decrease and an increase of fluorescence polarization mean an increase and a decrease of membrane fluidity, respectively. When comparing the interactivity between DPPC and neuro-mimetic membranes, the polarization changes (%) relative to control polarization values were used because the polarization values of intact membranes vary according to a difference in membrane lipid composition.

### 4.4. Estimation of Antagonism between Capsaicin and Capsazepine

For characterizing the property of capsazepine as a membrane-interacting antagonist, neuro-mimetic membranes were treated with capsaicin at 0.5–250 μM in the absence or presence of 50 μM capsazepine. The concentrations of capsaicin and capsazepine were fixed following the preliminary experiments of their combination. After the reactions at 37 °C for 45 min, DPH fluorescence polarization was measured as described in Section 4.3. The polarization changes were expressed as a percentage of the maximum increase induced by 200 μM capsaicin. IC_50_ and pA2 were calculated according to Science GatewayCell Biology Protocols [43] and a report of Ishikawa et al. [44], respectively.

### 4.5. Statistical Analysis

The data were statistically analyzed by one-way ANOVA with a Bonferroni post-hoc comparison using SPSS version 22 (IBM Corp.; Chicago, IL, USA). All results are expressed as mean ± SEM (*n* = 8 for each experiment on membrane interaction and *n* = 5 for estimation of membrane interaction antagonism), and values of ** *p* < 0.01 were considered statistically significant.

## Figures and Tables

**Figure 1 ijms-23-03971-f001:**
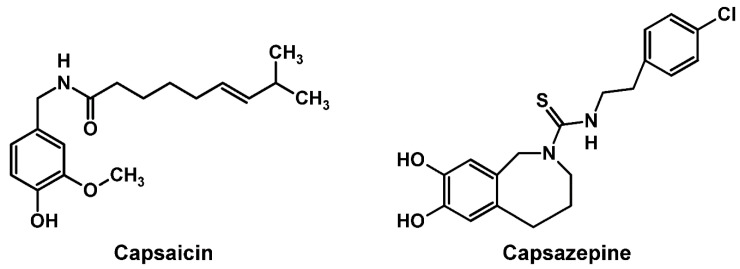
Structures of capsaicin and capsazepine.

**Figure 2 ijms-23-03971-f002:**
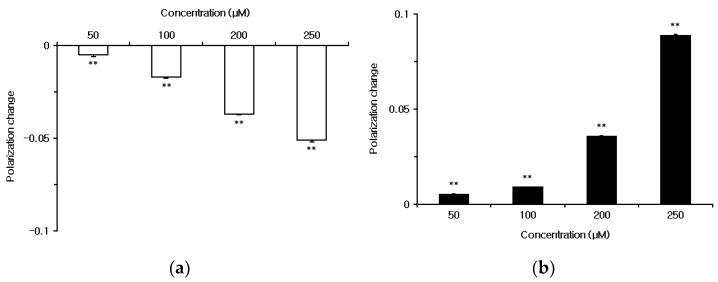
Interactions of (**a**) capsaicin and (**b**) capsazepine with 1,2-dipalmitoylphosphatidylcholine (DPPC) membranes. Each value represents mean ± standard error of the mean (SEM) (*n* = 8). ** *p* < 0.01 compared with controls.

**Figure 3 ijms-23-03971-f003:**
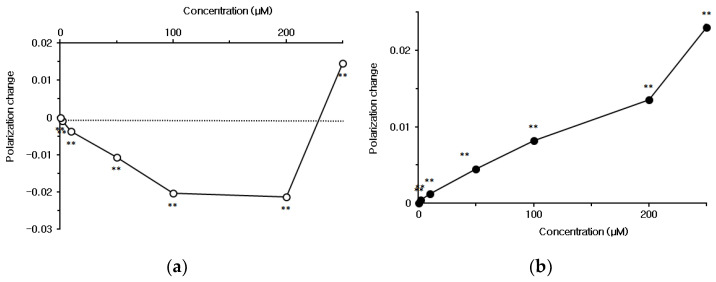
Interactions of (**a**) capsaicin and (**b**) capsazepine with neuro-mimetic membranes. Each value represents mean ± SEM (*n* = 8). ** *p* < 0.01 compared with controls.

**Figure 4 ijms-23-03971-f004:**
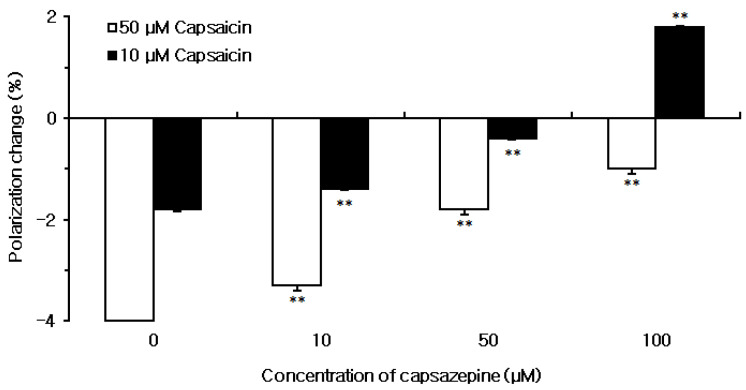
Effect of capsaicin on neuro-mimetic membranes and inhibition by capsazepine. Each value represents mean ± SEM (*n* = 8). ** *p* < 0.01 compared with capsaicin alone.

**Figure 5 ijms-23-03971-f005:**
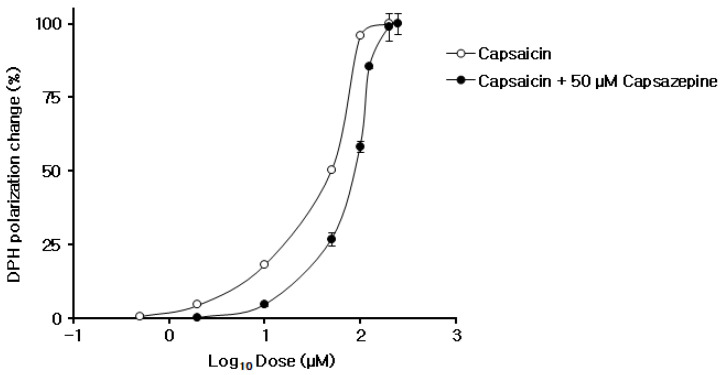
Effect of 50 μM capsazepine on DPH polarization changes of neuro-mimetic membranes induced by capsaicin at 0.5–200 μM. Doses (capsaicin concentrations) and responses (polarization changes) were plotted on the x- and y-axes, respectively, and the best-fit curve was plotted using the Hill equation. Open circles represent the changes by capsaicin in the absence of capsazepine and closed circles represent the changes by capsaicin in the presence of capsazepine. Each point represents mean ± SEM (*n* = 5).

**Table 1 ijms-23-03971-t001:** Comparison of membrane interactivity between DPPC and neuro-mimetic membranes.

Relative DPH Polarization Change (%)				
Concentration (μM)	Capsaicin		Capsazepine	
	DPPC Membrane	Neuro-Mimetic Membrane	DPPC Membrane	Neuro-Mimetic Membrane
50	−2.4 ± 0.5 **	−4.1 ± 0.0 **	2.6 ± 0.1 **	1.8 ± 0.1 **
100	−8.5 ± 0.4 **	−7.8 ± 0.0 **	4.5 ± 0.1 **	3.2 ± 0.1 **
200	−20.0 ± 0.2 **	−8.1 ± 0.0 **	17.9 ± 0.1 **	5.7 ± 0.1 **
250	−25.4 ± 0.5 **	5.5 ± 0.1 **	46.7 ± 0.3 **	9.7 ± 0.2 **

DPH, diphenyl-1,3,5-hexatriene. Each value represents mean ± SEM (*n* = 8). ** *p* < 0.01 compared with controls.

## Data Availability

Not applicable.

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
