# Peer review of "Membrane Interactivity of Capsaicin Antagonized by Capsazepine"

_ijms, 2022, doi:10.3390/ijms23073971_

Round 1

Reviewer 1 Report

Congratulations on your research and paper. 

Best regards 

Author Response

The authors greatly appreciate the comments.

Reviewer 2 Report

The authors report a very interesting study of the active role of lipid organization in the pharmacological activity of capsaicin. They not only check how this compound changes membrane fluidity, but also, and more interestingly, they tested the antagonism effect between capsaicin and a synthetic analogue. The methodology used is very well described and adequate, the results are clear and very relevant thus I recommend publication. However, the authors should address some comments before the paper can be published:

In the introduction the authors speak about the effect of amphiphiles in membrane organization. They should provide examples of how amphiphiles change for instance the phase transition of lipid membranes and, in particular, amphiphiles that serve as model anesthetics such as alcohols. The following references on effect of amphiphiles and active role of membrane organization should be added:

Shova Neupane, George Cordoyiannis, Frank Uwe Renner, Patricia Losada-Pérez, Real-time monitoring of interactions between solid-supported lipid vesicle layers and short-and medium-chain length alcohols: ethanol and 1-pentanol, Biomimetics 4, 8 (2019).

Thomas Heimburg, Andrew D. Jackson, The Thermodynamics of General Anesthesia, Biophysical Journal 92, 3159 (2007).

What is the size of the vesicles under study? Are they multilamellar or unilamellar? Can this influence interaction of the analgesic molecules with the membrane?

The authors explain very well and provide reported evidence of the molecular mechanism of capsaicin to increase membrane fluidity of DPPC as a function of concentration (Fig. 1A). In turn, capsazepine decreases membrane fluidity (figure 2 A). What is the molecular mechanism behind this increase? how does capsazepine accomodate within the membrane to decrease the fluidity? Does this mean that it increases the main phase transition temperature of DPPC?

The choice of lipid composition for neuro-mimetic membranes should be better motivated? Do these type of membranes exhibit phase separation or domain formation?

Reviewer 3 Report

In this paper, the authors explore the effects of capsaicin and capsazepine (TRPV1 channel agonist and antagonist, respectively) on neuro-mimetic membranes. They demonstrate that both compounds modify membrane properties depending on membrane lipid composition and have opposite effects on the fluidity of neuro-mimetic membranes.

The fact that many of the effects of amphiphilic phytochemicals are due to cell membrane perturbations, rather than specific protein binding, has long been known and well described in the literature. Although the information that capsaicin alters the properties of the lipid bilayer and the function of various membrane proteins is not new, this work provides new information on the interaction of capsaicin with capsazepine.

Methodologically, this is a straightforward study, carefully conducted and written in a concise style. The experiments are logically sequenced and performed with evident competence. I have two comments regarding the experimental design and interpretation and several minor suggestions below. 

Major:

  • Introduction, second paragraph (page 1-2, lines 42-51) should be rewritten and clarified because the analgesic effects of capsaicin are not „exclusively“ attributed to TRPV1 in the literature. Also, the effects of CZP other than on TRPV1 are well described.
  • The results of the experiments shown in Figure 5 (Page 4, lines 119-128) are not well explained. How many replicates were performed? How was the IC50 estimated? On what basis were the points shown in the graph connected by a curve (theoretical function?)? Are these results based on only one experiment? The effects do not seem to be purely competitive – this is not discussed.
  • Part of the discussion (Page 7, lines 230-252) is presented too extensively and is peripheral to the results presented. I recommend that this section be shortened considerably and only the relation to the presented results be emphasized…..
  • … Instead, the hypothesis proposed by the authors for a mechanistic explanation of the membrane interaction between capsaicin and CZP should be discussed. A molecular model of the action of capsaicin and CZP on the bilayer would greatly improve the study (e.g. see ACS Chem. Biol. 2014, 9, 8, 1788–1798).
  • In the discussion (Page 5, lines 145-150, and Page 6, lines 189-197), the authors mention in vivo experiments and consider possible systemic (non-proteinous) effects of higher concentrations of capsaicin and CZP through changes in membrane fluidity. However, I lack a discussion that relates the presented results to functional studies directly on the recombinant TRPV1 receptor - for example, it has been shown that TRPV1 channels desensitized by 1microM capsaicin can be fully reactivated by using a supersaturated concentration of capsaicin (30 microM) (Neuroscience 149, 2007, 144-154). Could this be explained by the effect of the high concentration of capsaicin on the membrane surrounding the receptor?
  • Minor:
  • Line 41, “local anesthetics” instead of “lical anesthetics”
  • Line 49, “..bilayer” instead of “bilasyer”
  • Line 179, “..capsaicin” instead of “caspaicin”

Round 2

Reviewer 3 Report

I thank the authors for responding to my comments and for correcting the manuscript.

However, I still have some concerns with the presentation of the competitive curve in Figure 5. Now that the points are the result of 5 repetitions, I find it strange that the points show no variance at all. Yet some variation in polarization changes is evident in the case of other measurements with 8 replicates. Otherwise, the legend must state that the error bars are smaller than the symbols. Could the authors provide (in their response to this reviewer) the original data from which the curves were constructed? Also, the legend of the Figure 5 must indicate how the points were connected by the curves.

Minor: I also noticed an additional error in the y-axis labeling in Figure 4. Please correct.

Author Response

Author’s Responses to the Comments of Reviewer 3

I thank the authors for responding to my comments and for correcting the manuscript.

The authors greatly appreciate reviewer’s suggestive comments.

However, I still have some concerns with the presentation of the competitive curve in Figure 5. Now that the points are the result of 5 repetitions, I find it strange that the points show no variance at all. Yet some variation in polarization changes is evident in the case of other measurements with 8 replicates. (1) Otherwise, the legend must state that the error bars are smaller than the symbols. (2) Could the authors provide (in their response to this reviewer) the original data from which the curves were constructed? Also, (3) the legend of the Figure 5 must indicate how the points were connected by the curves.

(1) The authors recalculated and found some errors. So, Figure 5 was remade after their correction (please see Page 5 of the revised MS).

(2) The examples of experimental data have been attached (please see the attachment). We think showing them in the text is not necessarily required.

(3) According to the comment, the explanation was added to the legend of Figure 5 (please see lines 134-136 in the revised MS).

Doses (capsaicin concentrations) and responses (polarization changes) were plotted on the x- and y-axes, respectively, and the best-fit curve was plotted using the Hill equation with reference to a study of Di Veroli et al. [An automated fitting procedure and software for dose-response curves with multiphasic features. Sci. Rep. 2015, 5, 14701.].

Minor: I also noticed an additional (4) error in the y-axis labeling in Figure 4. Please correct.

The original Figure 4 was replaced with the new one remade after correction of the y-axis labeling (please see Page 4 of the revised MS).
